# Current Challenges in the Diagnosis of Pediatric Cutaneous Mastocytosis

**DOI:** 10.3390/diagnostics13233583

**Published:** 2023-12-01

**Authors:** Hanna Ługowska-Umer, Justyna Czarny, Agnieszka Rydz, Roman J. Nowicki, Magdalena Lange

**Affiliations:** 1Department of Dermatology, Venereology and Allergology, Medical University of Gdańsk, 80-211 Gdańsk, Poland; hannaumer@gumed.edu.pl (H.Ł.-U.); justyna.czarny@gumed.edu.pl (J.C.); rnowicki@gumed.edu.pl (R.J.N.); 2Student’s Scientific Circle Practical and Experimental Dermatology, Medical University of Gdansk, 80-211 Gdańsk, Poland; agnieszka.rydz@gumed.edu.pl

**Keywords:** cutaneous mastocytosis, children, tryptase, KIT mutation, diagnostics

## Abstract

Pediatric mastocytosis is mostly a cutaneous disease classified as cutaneous mastocytosis (CM), which is characterized by mast cell (MCs) accumulation in the skin and the absence of extracutaneous involvement. Based on the morphology of skin lesions, CM can be divided into three major forms: maculopapular CM (MPCM), diffuse CM (DCM) and mastocytoma of the skin. A positive Darier’s sign is pathognomonic for all forms of CM. MPCM is the most common form, presenting with red-brown macules or slightly raised papules. Mastocytoma is characterized by solitary or a maximum of three nodular or plaque lesions. DCM is a rare, severe form which presents as erythroderma, pachydermia and blistering in the infantile period of the disease. CM is associated with MC mediator-related symptoms, most commonly including pruritus, flushing, blistering, diarrhea and cramping. Anaphylactic shock occurs rarely, mainly in patients with extensive skin lesions and a significantly elevated serum tryptase level. Childhood-onset MPCM and mastocytoma are usually benign diseases, associated with a tendency for spontaneous regression, while DCM is associated with severe mediator-related symptoms, an increased risk of anaphylaxis and, in some cases, underlying systemic mastocytosis (SM). In contrast to adults, SM is a rare finding in children, most commonly presenting as indolent SM. However, advanced SM sporadically occurs.

## 1. Introduction

Mastocytosis is a heterogenous group of neoplasms characterized by an expansion and accumulation of neoplastic mast cells (MCs) in the skin and/or internal organs, mostly the bone marrow (BM), spleen, lymph nodes, liver, and gastrointestinal organs [1,2,3]. The estimated prevalence of CM is 1–3 per 10,000, while the prevalence of SM is approximately 1 per 10,000 [1]. Based on the clinical presentation, mastocytosis can be divided into three major forms: cutaneous mastocytosis (CM), which is a skin-limited disease; systemic mastocytosis (SM), with BM and/or internal organs involvement; and mast cell sarcoma (MCS), presenting as an extracutaneous aggressive tumor [1,2,4]. An updated classification of mastocytosis, based on both the WHO and the European Union/United States consensus group statements, is presented in Table 1 [1,4,5].

Generally, the majority of pediatric patients suffer from CM, associated with a tendency for spontaneous regression around puberty and a favorable prognosis [6,7,8,9,10]. The majority of children experience disease onset within the first 2 years of life [6,7,9]. According to the international consensus statement, CM is divided into maculopapular cutaneous mastocytosis (MPCM), with two variants, namely monomorphic and polymorphic, diffuse cutaneous mastocytosis (DCM), and a cutaneous mastocytoma [5,6].

SM rarely occurs in children and mostly presents as an indolent SM (ISM) [7,8,9,11,12]. A recent update of the diagnostic criteria, established by the consensus group, included one major and three minor criteria [3]. A major criterion is defined as multifocal dense infiltrates of MCs (>15 MCs in aggregates) in BM and/or in sections of other extracutaneous organ(s), while minor criteria include (a) >25% of all MCs being atypical cells type I or type II on BM smears or spindle-shaped in MC infiltrates detected in sections of BM or other extracutaneous organ(s); (b) *KIT*-activating point mutation(s) at codon 816 or in other critical regions of *KIT* in BM or another extracutaneous organ; (c) MCs in BM, blood, or other extracutaneous organs express CD2 and/or CD25 and/or CD30; and (d) baseline serum tryptase concentration >20 ng/mL (in the case of an unrelated myeloid neoplasm tryptase does not count as an SM criterion). When hereditary alpha tryptasemia (HαT) is also present, the tryptase level should be adjusted [1,3]. If at least one major and one minor (or three minor) criteria are fulfilled, the diagnosis of SM is established [1,2,3]. Advanced forms of SM such as aggressive systemic mastocytosis (ASM), SM with an associated hematologic neoplasm (SM-AHN) and mast cell leukemia (MCL) are reported occasionally in children [11,12,13,14,15]. The diagnosis of various forms of SM is based on the presence or absence of B and C findings described elsewhere [1,2,3].

Regarding the pathogenesis of mastocytosis, somatic, gain-of-function point mutations within *KIT* play crucial role in the development of the disease [1,2,3,16,17,18]. KIT (CD117) is a type III receptor tyrosine kinase expressed by MCs. The interaction between KIT and its ligand stem cell factor (SCF) drives MC differentiation, maturation, adhesion, chemotaxis and survival [19], while the gain-of-function somatic mutations in the *KIT* tyrosine kinase domain leads to dysregulation of these processes, which results in the enhanced growth and accumulation of MCs in various tissues [1,2,3]. Both childhood-onset and adult-onset mastocytosis are clonal in nature [16,17,20,21,22]. It was found that the D816V *KIT* mutation in exon 17 is most common in adults with SM, in whom this mutation was detected in over 90% of cases [16,23]. In contrast, in children, the D816V *KIT* mutation was detected in lesional skin only in 36% of patients [17]. Other *KIT* mutations, mainly affecting exons 8, 9, 10 or 11, were reported in approximately 40% of pediatric patients, whereas 25% had no *KIT* mutations (*KIT*-wild type) [17,18,20,24]. Moreover, no evident correlation between *KIT* mutations and clinical phenotype and the prognosis of pediatric mastocytosis was found [17,18,20]. In some cases of DCM, various somatic *KIT* mutations (D816V, D816Y, D816I, Del419, K509I, internal tandem duplication A502_Y503dup) have been detected [17,20,21,24,25,26,27]. In recent years, identifying and quantitating the *KIT* D816V mutation in peripheral blood (PB) became a reliable predictor of SM in adults [28]. A similar approach was taken regarding pediatric patients; PB ASqPCR for the *KIT* D816V mutation was negative in children with CM, whereas it was positive in the majority of patients with ISM [29]. This non-invasive examination is a useful diagnostic tool in children with elevated serum tryptase levels and suspected SM [29,30]. Generally, pediatric mastocytosis is not a hereditary disease associated with germline mutations of *KIT* [7,9,16]. In rare cases of familial DCM, germline mutations such as S451C, A533D and the *KIT* variant c.1598C>A (p.Ala533Asp) were detected [31,32,33]. Occasionally, mastocytosis associated with tuberous sclerosis and gastrointestinal stromal tumors, with germline or somatic *KIT* mutations, has been reported [34,35,36,37]. Despite numerous studies, the precise role of all genetic alterations in the pathogenesis of pediatric mastocytosis is not fully understood.

## 2. Clinical Characteristic of Skin Involvement in Children with CM

The majority of children with mastocytosis purely have CM. In 2007, the European Union/US consensus group established criteria for cutaneous involvement in patients with mastocytosis, including the presence of typical skin lesions with a positive Darier’s sign (major criterion) and one or two of the following minor criteria: histologically confirmed infiltration of MCs in the dermis (around 40 MCs/mm^2^) and activating *KIT* mutations at codon 816 in lesional skin [38]. The Darier’s sign, pathognomonic for CM, presents as a wheal-and-flare reaction of the lesion which occurs within a few minutes after stroking a CM lesion around 5 times, using moderate pressure, with a tongue spatula [6] (Figure 1). All three subforms of CM, including MPCM with polymorphic and monomorphic variants, DCM and mastocytoma, were precisely characterized in the consensus report published in 2016 [6]. A review of the literature data on childhood-onset mastocytosis shows that the clinical phenotype of skin lesions is heterogeneous [6,9,39,40,41,42,43].

Polymorphic MPCM is the most common form of CM in children [6,7]. It is characterized by brown to red oval lesions, plaques and nodules of different sizes, distributed asymmetrically on the skin (Figure 2). The size of said lesions is bigger compared to cutaneous lesions in adult-onset mastocytosis [6,9,39]. The typical location of the lesions is the head, particularly the lateral parts of the forehead, the neck and the extremities [6]. Interestingly, the morphology of skin lesions may change during the course of the disease [6,9]. Blistering is common particularly in infancy and ceases when the patient is 2 or 3 years of age [6,7,9]. Nodular lesions, if present in infancy, may evolve into plaques at the age of 5–10 years and then regress around puberty or flatten with time and mimic anetoderma [6]. The majority of patients with polymorphic lesions have a skin-limited disease [6,7,9]. Serum tryptase levels are usually in the normal range, except for children with extensive skin lesions [42,44,45,46]. Children with MPCM usually suffer from MC mediator-related symptoms, which are more prominent in those with extensive skin lesions [47,48,49,50]. Polymorphic MPCM is associated with a favorable prognosis of a spontaneous regression of skin lesions by adolescence [6,51]. Children with this cutaneous manifestation of the disease usually have a normal serum tryptase level and a tendency towards regression after puberty [51]. Some children may present atypical CM with yellow papules or nodules resembling xanthomas, which represent an atypical, rare variant termed xanthelasmoid CM [6] (Figure 3).

Monomorphic MPCM, commonly observed in adults, is less prevalent in the pediatric population [6,7,9]. In most patients, disease onset is after the age of 24 months [51]. Monomorphic MPCM is characterized by small brownish maculopapular lesions of identical shape and size, located mainly on the trunk and thighs [6,7,39] (Figure 4). Blisters occurs rarely. The lesions present a lower tendency for spontaneous regression, may persist into adulthood, and represent a cutaneous manifestation of SM [6,9,51]. The serum tryptase levels vary from normal ranges to highly increased values [51]. 

DCM patients present with a generalized erythroderma and pachydermia (thickened skin) with blistering, as a result of MCs’ infiltration of the entire skin [6,26,41,43,52,53,54]. DCM is the rarest form of CM (1–5%) [7,12,55]. The onset of DCM is usually reported at birth or in early infancy [52,53,54,56]. Pronounced dermographism and Darier’s sign are observed. Scratching of the skin provokes a release of MC mediators and reddening of the skin, urticarial lesions and blisters [26,53,54]. Blister formation is present in the majority of DCM patients and may be the first manifestation of the disease [6,26,53,54,55,57]. The tendency for blister formation usually decreases after 2–3 years of life [7,9]. Extensive blistering, presenting as both small vesicular lesions and large hemorrhagic bullae, is a dominant clinical manifestation in infancy (Figure 5), while a leather-like appearance and hyperpigmentation (brown or yellow colour of the skin) develops in the further course of DCM [7,9,43,52,54,55]. DCM is associated with severe MC mediator-related symptoms and a higher risk of anaphylactic shock than in other forms of CM [39,42,48,50,54,58,59]. The large MC infiltration of the infant’s entire skin and an unfavorable ratio of the body weight to body surface area are the reasons for DCM patients usually presenting with elevated serum tryptase levels [7,9,26,42,53]. Due to the severe MC mediator-related symptoms, children with DCM often require hospitalization [9,48]. In the majority of children who lack systemic organ involvement, cutaneous lesions often resolve by adolescence [6]. Children with familial DCM usually have persistently increased tryptase levels, while a chronic course of the disease and mast cell infiltration in extracutaneous organs occur in some [31,32,33,34,51]. It is worth pointing out here that skin lesions corresponding to DCM may occur in children with SM, in whom the disease leads to fatal outcomes [25,60,61,62,63].

Mastocytoma presents as a solitary or a maximum of three elevated, brown or yellow lesion(s), 1–10 cm in diameter, with possible blisters in the infantile period of the disease [6,64] (Figure 6). A typical location of these lesions is the trunk. A Darier’s sign may be prominent and a blister after mechanical stimulation may appear [43,55,58,64,65]. The onset of the disease is usually observed before 6 months of age or even at birth [39,64,66]. The lesions regress spontaneously before puberty [6,7]. Serum tryptase levels are in the normal ranges and no systemic involvement has been reported [20].

## 3. Mast Cell Mediator-Related Symptoms and Anaphylaxis in Children with CM

Activation of the MCs in patients with mastocytosis leads to the degranulation of multiple mediators. Specific for immediate-type allergic reactions, MCs’ activation by high-affinity receptors for immunoglobulin E (IgE) is one of the important pathways. Moreover, multiple agents such as foods, drugs, radiocontrast media, certain cytokines, anaphylatoxines, neuropeptydes, immunoglobulin G (IgG) immune complexes, complement products of bacteria or parasites, as well as physical factors including heat, cold, stress, sun exposure and physical effort may provoke MCs’ activation and degranulation, and may also act as co-factors in allergic and anaphylactic reactions [47,67,68].

Performed mediators, stored in cytoplasmic granules, are responsible for the early phase of an allergic reaction. These are histamine, tryptase, chymase, carboxypeptidase, heparin, neutrophil chemotactic factor and eosinophil chemotactic factor. The other type of MC mediators are the newly formed mediators, namely prostaglandin D2, leukotriene C4, leukotriene D4, platelet activating factor and cytokines, such as tumor necrosis factor α (TNF α), interleukin-4 (IL-4), interleukin-5 (IL-5), and interleukin-6 (IL-6), responsible for the late phase of an allergic reaction [47,67]. Some of these cytokines were examined in murine MCs, and therefore their role in humans remains unclear. In humans, interleukin-8 (IL-8), monocyte chemoattractant protein-1 (MPC-1) and oncostatin-M, among others, may play a role in MC activation [47,69,70,71].

In children with CM, the most common are cutaneous MC mediator-related symptoms, including pruritus, flushing, blistering and dermographism, regardless of the subtype of CM [39,41,47,59]. Pruritus is the most commonly reported symptom [6]. In the largest systematic review of 1747 children with mastocytosis, pruritus was reported in 48% of patients, flushing was present in 24.5% of children, blisters were present in 34.5% of children and gastrointestinal symptoms were observed in 19.5% of patients [12]. The Darier’s sign was present in up to 90% of patients [6]. Blistering typically occurs in infants and children under 2–3 years of age, mostly in those with DCM, and is considered a predictor for severe complications [7,12,49,54,72]. Flushing, relatively common in CM children, manifests as a sudden redness of the skin and is usually present on the face, neck and, less frequently, on the upper trunk. Episodes of flushing occur in up to 30–50% of patients with MPCM, especially in those with extensive skin lesions, particularly in patients with DCM [54,58]. Flushing, in some cases, may prelude a syncope or even anaphylactic shock [48].

Extracutaneous MC mediator-related symptoms in children with mastocytosis are not very common. Children may suffer from gastrointestinal symptoms, mainly cramping, abdominal pain, diarrhea, nausea and vomiting [9,45,49,58]. Patients with moderate or severe symptoms should be referred to gastroenterologists for further investigation [47]. Respiratory symptoms, which may include bronchoconstriction, rhinorrhea, wheezing, stridor and cough, are seen in approximately 13% of children with mastocytisis [49]. Neurological symptoms, rarely observed in children with CM, include a lack of concentration, aggressive behavior, anxiety and depression episodes [47,73]. Children with extensive skin lesions and elevated serum tryptase levels may also suffer from cardiovascular symptoms, including tachycardia, hypotension, collapse and shock [9,48,49,54]. Constitutional MCs mediator-related symptoms, typical for SM, such as fever, fatigue, weight loss, musculoskeletal pain, osteopenia, and osteoporosis, are very rarely reported in children with mastocytosis [12,45].

A review of the literature data shows that anaphylaxis occurs in approximately 4% of children with mastocytosis, and is thus much higher than in the general pediatric population (0.02–0.05%) [47,74]. The risk factors of anaphylaxis include elevated serum tryptase level, extensive skin lesions, blistering, flushing, DCM, SM, previous episodes of anaphylaxis and hereditary alpha tryptasemia (HαT), which is a genetic trait [47,48,49,50,68,75]. Spanish authors reported that 12 out of 111 children suffering from mastocytosis required hospitalization due to severe, life-threatening MC activation symptoms [48]. All of those children presented with extensive skin lesions (over 90% of body surface area) and elevated serum tryptase levels [48]. Moreover, nine of them were diagnosed with DCM and only three with MPCM. Blistering also occurred as a risk factor for severe symptoms requiring hospitalization [48]. The results of these studies clearly indicate that the risk of anaphylactic reactions is higher in patients with elevated serum tryptase levels and extensive skin lesions with blistering episodes than in those with limited skin involvement [48].

Insect venom, the most potent trigger of anaphylactic shocks in adults with mastocytosis, is a very rare elicitor of anaphylaxis in pediatric patients [73,76,77]. Foods and drugs were reported in several studies as being triggers of anaphylaxis in children with mastocytosis [47,54,73,77]. Commonly used medications, such as nonsteroidal anti-inflammatory drugs (NSAIDs), are rarely associated with MC mediator release in children. Anesthetic procedures and vaccinations rarely provoke anaphylaxis in pediatric patients with mastocytosis [9,44,47,78]. Important triggers of MC degranulation are environmental factors such as exposure to sudden temperature changes, emotional stress, fever or intensive exercises [7]. However, in many cases, the trigger remains unknown; therefore, idiopathic anaphylaxis is commonly reported in pediatric patients with mastocytosis [7,45,47,75].

## 4. Diagnostic Evaluation in Children with Suspected Mastocytosis

In recent years, an ongoing debate on how to optimize the diagnostic workup in children with suspected mastocytosis has taken place. Numerous proposals regarding diagnostic algorithms have been published, reflecting a deeper understanding of the specificity of clinical manifestations and pathogenesis of pediatric mastocytosis over time [7,8,10,29,79]. Before making a decision on the diagnostic workup in an individual patient, it is worth considering several aspects. Firstly, the vast heterogeneity of childhood mastocytosis ranging from benign solitary cutaneous mastocytoma to advanced SM should be taken into account [7,11,12]. Secondly, serum tryptase may be elevated in children with very extensive skin lesions without underlying SM due to a heavy MC burden in the entire skin [41,42,45,46,47,79]. Thirdly, both mastocytosis and HαT may coexist and present with overlapping symptoms, as well as an elevated basal serum tryptase level, which makes the distinction between these conditions a difficult task [80,81,82,83,84]. It seems reasonable to advise screening for HαT in children with elevated serum tryptase levels before proceeding with a BM biopsy [85]. However, some authors recommend a genetic test for the *TPSAB1* copy number in all pediatric patients with a serum tryptase level higher than 6.5 ng/mL [8]. Crucially, the diagnosis of HαT in a child does not rule out the need for determining the activating *KIT* mutations in PB and/or a BM biopsy in patients in whom SM or another myeloid neoplasm is suspected. Naturally, more studies are needed to establish the role of tryptase genotyping in the diagnostic algorithm and differential diagnosis in pediatric mastocytosis [85]. Nevertheless, no public laboratories routinely offer a targeted genetic test for the *TPSAB1* copy number nowadays. Taking all of the above into account, each universal diagnostic algorithm seems to be unreliable in individual cases. In our experience with childhood-onset mastocytosis, personalized management, based on current knowledge, may be the best solution. Thus, herein we present a tabulation of various diagnostic procedures which should be considered in children with skin lesions suggestive for CM and those with the established diagnosis of CM (Table 2), as well as a brief description of the main diagnostic tools. As pediatric mastocytosis is mostly a cutaneous disease, the diagnosis of the majority of cases is based on the typical morphology of skin lesions, a positive Darier’s sign and the histology of lesional skin (if necessary) [6,7,10,41]. Moreover, in all of them, noninvasive screening examinations should be performed to exclude underlying SM. The most important procedures include the following: obtaining the medical history, performing a physical examination, performing a complete blood count with differentials, conducting serum chemistry analysis (hepatic function panel, albumin, renal function panel, beta2-microglobulin), measuring the basal serum tryptase level [8,10,40,44,47,68,86]. Moreover, performing an abdominal ultrasound is also indicated, as organomegaly is a strong predictor of SM in children.Children with significant abnormalities in the complete blood count, organomegaly and a significantly elevated and/or rising serum tryptase level or other signs and symptoms suggesting the presence of advanced SM, or another systemic hematologic neoplasm, first require the determination of the *KIT* D816V mutation in PB using the sensitive allele-specific quantitative polymerase chain reaction (ASqPCR) [7,29,30,79]. Those with only a moderately elevated serum tryptase level and/or numerous mediator-related symptoms should first undergo a screening for HαT. Of note here is that a significantly elevated serum tryptase level (>20 ng/mL) is not in itself an indication to perform BM studies [7,79]. Selected pediatric patients with rising *KIT* D816V variant allele frequency (VAF) in PB and suspected advanced SM should undergo a BM biopsy and aspiration as well as complete staging strictly according to the recommendations for adults with suspected SM described elsewhere [1,2,3]. The same workup is recommended in patients with childhood-onset mastocytosis in whom skin lesions persist into adulthood [7,40]. In the remaining children with cutaneous lesions of mastocytosis who have no signs and symptoms indicative of the presence of SM or another systemic hematologic neoplasm, the final diagnosis of CM can be established [7].

### 4.1. Skin Histology

A skin biopsy and a histology of lesional skin are not obligatory diagnostic procedures in all children with cutaneous lesions of mastocytosis [7,8,10]. The diagnosis of CM may be established based on clinical features, but only in children presenting with typical lesions corresponding to MPCM, DCM and mastocytoma, as well as a positive Darier’s sign [6,7]. It is worth noting here that Darier’s sign is only pathognomonic if strongly positive, which can be observed when dense MC aggregates are present in the skin [87]. In all clinically unclear cases, particularly if MPCM lesions are discrete and Darier’s sign is non-diagnostic or negative, a confirmation of diagnosis via skin biopsy is required [10,40,41,87]. The routine process is to obtain the biopsy using a 4 mm punch biopsy device after administrating local anesthesia. The use of antibodies against tryptase and/or CD117 is a recommended immunohistochemical marker of MCs to evaluate and quantify mast cells in skin biopsies [10,38]. Staining for CD2 and CD25 is usually negative in cutaneous MCs, whereas CD30 staining is frequently positive in lesional MCs in children with MPCM and mastocytoma [87,88,89]. Importantly, there was no correlation between the immunophenotype of MCs in the lesional skin and the form of CM or the disease course in pediatric mastocytosis patients [88,89]. MCs in lesional skin are either spindle-shaped or round. Monomorphic MPCM usually has a lower density of MCs with a spindle shape, whereas a high density of MC with a round or cuboidal shape (spherical) is more typically observed in polymorphic MPCM lesions [6,58]. Recently, the histopathological criteria of CM were validated by Gebhard et al. and published in 2022 [87]. The major histological criterion for the diagnosis of MPCM with ≥95% specificity is MC density in the upper dermis > 139 MC/mm^2^ (equivalent to about 27 MC/HPF) [87]. Apart from that, the following minor criteria were established: MC density 8–12/HPF (specificity, 86.1%; sensitivity, 68.8%), strong/intermediate basal pigmentation (specificity, 50%; sensitivity, 96.9%), the presence of MC clusters with more than cells with nuclei (specificity, 100%; sensitivity, 6.3%) and the presence of interstitial MC (specificity, 50%; sensitivity, 96.9%) [87]. Moreover, based on all histological criteria, a scoring model to predict MPCM was designed [87]. The following grading system was used: MC density ≥ 27 MC/HPF (a score of 3), MC density ≥ 12 MC/HPF (a score of 2), MC density ≥ 7 MC/HPF (a score of 1), strong/intermediate basal pigmentation (a score of 1), interstitial MC (a score of 1) and MC clusters with more than 3 cells with nuclei (score 1). Scores of 4–6 confirm the diagnosis of CM. Interestingly, it was also found that MCs stained positive with tryptase, CD117 and CD45 were spindle-shaped (skin biopsies obtained from 32 MPCM, 15 mastocytoma) and negative for CD30 in all cases [87]. MC density decreased from the epidermal layer to the deeper layers in MPCM patients. Furthermore, MCs were more often located interstitially compared to the controls, and their size was larger than in controls [87]. Recently, differences in histopathological features between monomorphic MPCM and polymorphic MPCM or DCM were also examined to facilitate a precise diagnosis of the form of CM [90]. It was found that monomorphic MPCM displayed a distinct pattern of MC infiltration, in which MC infiltration is mainly situated in the reticular dermis with sparing of the papillary dermis, whereas in polymorphic MPCM and DCM, dense MC infiltration in the papillary dermis is observed [90]. Moreover, the same study revealed similarities in the histopathological picture of childhood-onset and adult-onset monomorphic MPCM [90]. These results suggest that the lack of MC infiltration of the papillary dermis may help to exclude monomorphic MPCM when the clinical phenotype is not clear. It is worth noting that in patients with DCM and mastocytomas, a higher burden of MCs in the skin tissue is observed than in those with MPCM [6,87].

### 4.2. Molecular Testing for Activating KIT Mutations in the Skin and Peripheral Blood 

The determination of the *KIT* mutational status in the skin is an important diagnostic criterion for CM; however, it is not a necessary procedure in children with CM confirmed by means of histology in daily practice. Genetic examination is of particular importance in all unclear cases, as it is the most specific criterion for cutaneous lesions of mastocytosis. It is recommended to determine the *KIT* D816V mutational status with a highly sensitive quantitative real-time PCR [87,91]. Recently, it was shown that in formalin-fixed skin samples with MPCM, processed in the routine setting, the *KIT* D816V mutation may be detected with 100% specificity and 100% sensitivity [87]. The presence of the *KIT* D816V mutation in FFPE (formalin fixed paraffin embedded) and a MC density > 27/HPF are over 95%-specific major criteria for MPCM. The determination of the *KIT* mutational status in the skin of children with DCM is also indicated in selected cases when therapy with tyrosine kinase inhibitors is considered [27]. It is worth mentioning that the correlation between *KIT* mutations in the skin and both the phenotype of childhood-onset mastocytosis and the clinical outcome is not clearly established [12,18,20,21,79]. One should also keep in mind that the detection of a *KIT* D816V mutation in the skin is not a diagnostic criterion of SM. 

As mentioned above, the determination of the *KIT* D816V mutation (ASqPCR) in PB has recently been recommended in children with suspected SM before the decision to perform a BM biopsy is considered [28,29,79]. Identifying the subgroup of patients who need more careful follow-up and further BM studies in selected cases may be helpful [7,29,30,79]. This non-invasive examination is a standard procedure in children with organomegaly (hepatosplenomegaly, lymphadenopathy) and/or a significantly elevated and/or rising serum tryptase level (>20 ng/mL) and/or significant abnormalities in PB [7,29,79]. Moreover, it may also be considered in patients with severe MC mediator-related symptoms, anaphylaxis and extensive skin involvement [29,30]. Patients with symptoms suggesting systemic disease and a negative *KIT* D816V mutation should be evaluated for other *KIT* mutations [10,29,79]. It is notable that a positive result for the *KIT* D816V mutation in PB, with the absence of other clinical symptoms, does not determine the diagnosis of SM, as it is only one of the minor criteria of SM [2,3]. Both the presence of *KIT* D816V (ASqPCR) in PB and organomegaly are the most reliable predictors of SM in children [7,10,29,48,79].

### 4.3. Serum Tryptase Level

Tryptase is a member of the serine protease family, primarily synthesized and stored in the MCs [92]. The basal serum tryptase level reflects the body’s total MC burden and, therefore, is employed in the diagnostic workup for suspected SM, as well as follow-up [38]. Constitutive basal serum tryptase level is relatively stable in humans, but it may vary depending on genetic features, comorbidities and other factors [92]. Importantly, HαT may be a reason for an elevated basal serum level [82,93]. Throughout the human lifetime, the level of serum tryptase in an individual person may change due to renal failure or other comorbidities such as chronic inflammatory diseases, chronic infections and associated hematological diseases [94]. Moreover, the serum tryptase level increases transiently during anaphylactic reactions [95]. Therefore, basal serum tryptase level should be determined 24–48 h following the clinical resolution of anaphylaxis [38]. The median serum tryptase level in a healthy population is approximately 5 ng/mL [38]. Recently, the normal range of baseline serum tryptase level between 1 ng/mL and 15 ng/mL was established by Europeans Competence Network on Mastocytosis (ECNM) and the American Initiative on Mast Cell Disease [92]. Elevated serum tryptase levels are found in the majority of patients with SM [51,92]. A basal serum tryptase level of >20 ng/mL is defined by WHO as a minor diagnostic criterion of SM [2]. In childhood-onset mastocytosis, elevated serum tryptase levels are observed in patients with extensive skin lesions, particularly in those with DCM [45,46]. Patients with monomorphic MPCM have varied levels, ranging from normal to highly increased values, whereas children with polymorphic MPCM and mastocytoma usually present levels within the normal range [45,51]. 

### 4.4. Screening for Hereditary Alpha Tryptasemia

Until now, little was known about the frequency and symptomatology of HαT in children with mastocytosis [80,83,84,85,96]. HαT is a genetic trait characterized by an increase in the copy number of the alpha tryptase gene *TPSAB1*, resulting in elevated basal serum tryptase levels and an increased risk of severe hypersensitivity reactions [80,81,83,97]. In patients with SM, concomitant HαT is considered an additional risk factor of anaphylaxis [83,97]. Importantly, approximately two-thirds of individuals with HαT show minimal or no symptoms, whereas the remaining individuals may present with various symptoms, including Hymenoptera venom allergy, flushing, pruritus, urticarial/angioedema, irritable bowel syndrome, gastroesophageal reflux, arthralgia, hypermobility, inhalant allergies, asthma, sleep disruption, neuropsychiatric symptoms and dysautonomia [81,83]. In children with purely CM anaphylaxis, Hymenoptera venom allergy and elevated serum tryptase are rare findings [42,45,46,47]. Therefore, it is controversial as to whether the analysis of *TPSAB1* copy numbers in children with CM should be recommended as a part of standard examination in this population [8,47,85]. However, it should be considered in children with basal serum tryptase ≥8–10 ng/mL, SM and severe MC mediator-related symptoms [47,85]. In children with elevated basal serum tryptase, non-invasive screening for HαT may help to stratify the risk of anaphylaxis and identify those who really require a BM biopsy. Tryptase genotyping is mainly useful in children with persistently elevated basal serum tryptase who do not present with other clinical features indicating SM, such as organomegaly or significant abnormalities in PB. One should also keep in mind that both HαT and mastocytosis may coexist and that the diagnosis of HαT does not rule out SM or other myeloid diseases [82,84]. According to recently updated diagnostic criteria for SM, when HαT is a concomitant diagnosis, the serum tryptase level should be adjusted [1]. A droplet digital polymerase chain reaction (ddPCR) assay is employed to demonstrate an increased copy number of the *TPSAB1* gene responsible for encoding alpha-tryptase [98]. Nevertheless, testing for *TPSAB1* copy numbers is not yet available in all centers.

## 5. Differential Diagnosis in Children with Suspected CM

### 5.1. Differential Diagnosis of MPCM

The long-lasting persistence of maculopapular rush, the brown colour of lesions and a positive Darier’s sign allow us to distinguish MPCM from urticaria, in which skin hives persists for up to 24 h and disappear without hyperpigmentation. Differential diagnosis of MPCM also includes drug eruption, post-inflammatory hyperpigmentation, ashy dermatosis, piloleiomyomas, lichen planus pigmentosus, Langerhans cell histiocytosis (LCH) and disseminated juvenile xanthogranuloma [43]. Interestingly, piloleiomyomas may have a clinical manifestation similar to MPCM, because they sometimes present with a pseudo-Darier’s sign, may present as a solitary or multiple nodules or multiple dermal nodules, and the colour of lesions ranges from pink to red, brown or skin-coloured [99,100]. A characteristic feature of piloleiomyomas is pain, which may appear spontaneously or as a result of pressure, low temperature, strong emotions or a light touch [99,100]. In CM, the same triggers may cause itching, blistering, flushing or Darier’s sign but pain is unlikely. LCH with a skin manifestation in infancy may also be taken into consideration because of the wide spectrum of lesions, including blisters, brown/purple papules and crusted nodules [101]. The eruption may be extensive and involve the scalp, face, trunk, buttocks and intertriginous area. To establish the diagnosis of LCH, it is necessary to perform a histological examination with immunophenotyping, revealing the presence of markers such as CD1a and/or CD207 [101]. Blistering in infants and small children with MPCM may be similar to bullous arthropod bites or a bacterial disease like bullous impetigo (a *Staphylococcus aureus*, group II infection, where erosions are covered by honey-yellow colour crusts); therefore, it may be taken into consideration in differential diagnosis [102]. Furthermore, in MPCM cases with spread bullae, autoimmune bullous diseases, especially linear immunoglobulin A bullous dermatosis (LABD), should be considered [103]. In LABD, tense small bullae on an inflammatory base, with an annular configuration and central crusting, are observed. Direct immunofluorescence examination confirms this diagnosis, revealing linear immunoglobulin deposition, primarily immunoglobulin A, at the dermoepidermal junction [43,102,103,104]. 

### 5.2. Differential Diagnosis of DCM

Bullous lesions may occur in all clinical forms of CM; however, in DCM, the blisters may be extensive, large and sometimes hemorrhagic [54]. For this reason, the differential diagnosis of DCM mainly includes bullous diseases, including Staphylococcal scaled skin syndrome (SSSS), LABD, epidermolysis bullosa, bullous impetigo and bullous erythema multiforme [43,54,102,105,106]. SSSS is a disease characterized by denudation of the skin caused by exotoxins produced by the *Staphylococcus* species. It usually occurs 48 h after birth and is rare in children older than six years. The disorder is characterized by significant exfoliation of the skin. The severity may vary from a few blisters to exfoliation of the entire skin, leading to marked hypothermia and hemodynamic instability [102,105,106]. The differentiation of DCM from epidermolysis bullosa, bullous erythema multiforme and bullous impetigo (if bullous lesions are limited) is of great importance in infants [12,53,55,106,107,108]. Epidermolysis bullosa congenita is an inherited disease in which blisters are often present at birth. Apart from blisters, children suffer from dystrophic nails, milla and atrophy of the skin. Diagnosis is confirmed by the presence of typical gene mutations characteristic for each subtype of the disease (simplex, junctional, dystrophic). Bullous erythema multiforme is an acute, immune-mediated disease triggered by an infection and/or drugs. Typically, it presents with target erythematous lesions with well-defined borders. The center of these lesions is covered by a blister. Insidious onset and infectious triggers, as well as coexisting systemic symptoms, such as fever, are a basis for the diagnosis [12,53,55,106,107,108]. In neonates and infants with DCM presenting with entire skin involvement, erythoderma and pachydermia, both severe atopic dermatitis and congenital disorders, such as severe combined immunodeficiency (SCID), should be considered [12,53,55,106,107,108]. As presented above, establishing a diagnosis based on the clinical presentation may be challenging in DCM children, particularly because of the absence of maculopapular lesions typical for CM and the rarity of DCM [54].

### 5.3. Differential Diagnosis of Mastocytoma

When it comes to multiple mastocytoma, Darier’s sign is helpful to distinguish these lesions from other diseases; however, in the case of a solitary cutaneous mastocytoma, a positive Darier’s sign was reported in half of the cases [64]. In all unclear cases, histology is crucial. Differential diagnoses of cutanous mastocytoma include Spitz nevus, congenital melanocytic nevus, juvenile xanthogranuloma, benign cephalic histiocytosis, café-au-lait macule, amelanotic melanoma, neurofibromatosis type I and granuloma faciale [43,64,65,101,109]. When cutaneous mastocytoma is a macule with non-diagnostic Darier’s sign, the differential diagnosis should consider café-au-lait macule, post-inflammatory hyperpigmentation or granuloma faciale [64,65,109]. Granuloma faciale is a rare skin disorder, characterized by single or multiple papules, plaques or nodules, most often occurring on the face. These lesions are usually varied in color (skin-colored or reddish-brown), size (from a few millimeters to several centimeters) and are elevated and soft in touch. Diagnosis is confirmed via a skin biopsy in which the inflammatory cells in the dermis include eosinophils; features of vasculitis are also present [109]. In cases with pruritus, differential diagnoses include a nodular scabies or an arthropod bite, while, when blistering is observed, cutaneous mastocytoma(s) may be distinguished from herpes simplex, bullus impetigo and bullous scabies [64,65,109]. 

## 6. Final Remarks

The review of the literature data confirms that in the majority of children, mastocytosis is a skin-limited disease, commonly associated with the presence of MC mediator-related symptoms. Currently, the diagnostics of CM has been facilitated as the histopathological criteria of MPCM have been validated [87]. Moreover, it has been shown that the presence of the *KIT* D816V mutation in FFPE skin confirms the diagnosis of MPCM in unclear cases. The most frequent form of CM in the pediatric population is polymorphic MPCM, in contrast to adults, who mainly present with monomorphic MPCM. Generally, the prognosis of childhood-onset mastocytosis is considered as good, particularly in children with mastocytoma and polymorphic MPCM [76,110,111,112]. Recent studies confirm the tendency towards the spontaneous regression of skin lesions around puberty; however, the complete regression is not as often evaluated [11,20]. A late onset of the disease (after 2 years), as well as monomorphic MPCM, is associated with a longer course of the disease and a lower tendency for regression [6,20,51]. DCM, a very rare and the most severe form of CM, is usually associated with intense MC mediator-related symptoms, particularly extensive blistering in infancy and an increased risk of anaphylaxis. In the majority of cases, DCM tends to resolve itself; nevertheless, in some cases, children with cutaneous lesions corresponding to DCM present with SM and have a fatal outcome. Anaphylaxis occurs rarely in childhood-onset mastocytosis; the risk factors include an elevated serum tryptase level, DCM, extensive skin lesions, severe blistering and flushing, HαT and SM. Basal serum tryptase, usually in normal ranges in children with mastocytoma and MPCM and elevated in children with extensive skin lesions, is a useful diagnostic tool in pediatric mastocytosis. Moderately elevated basal serum tryptase raises a suspicion of HαT, whereas tryptase levels which are significantly elevated and rising over time suggest the presence of SM, which may occur in children. Currently proposed diagnostic algorithms include screening for HαT and determination of activating *KIT* mutations in PB [7,8,10,29,79]. Positive *KIT* D816V in PB and organomegaly are considered the strongest predictors of SM. Until now, significant correlations between genetic status, as well as the immunophenotype of MCs in the skin, the disease phenotype and prognostication in pediatric mastocytosis have not been established [18,20,88,89]. Multicenter, large-scale and long-term follow-up studies on childhood-onset mastocytosis are still required to define the risk factors of SM, anaphylactic shock and factors associated with the regression of skin lesions around puberty, in order to improve the management and better predict the outcome of the disease.

## Figures and Tables

**Figure 1 diagnostics-13-03583-f001:**
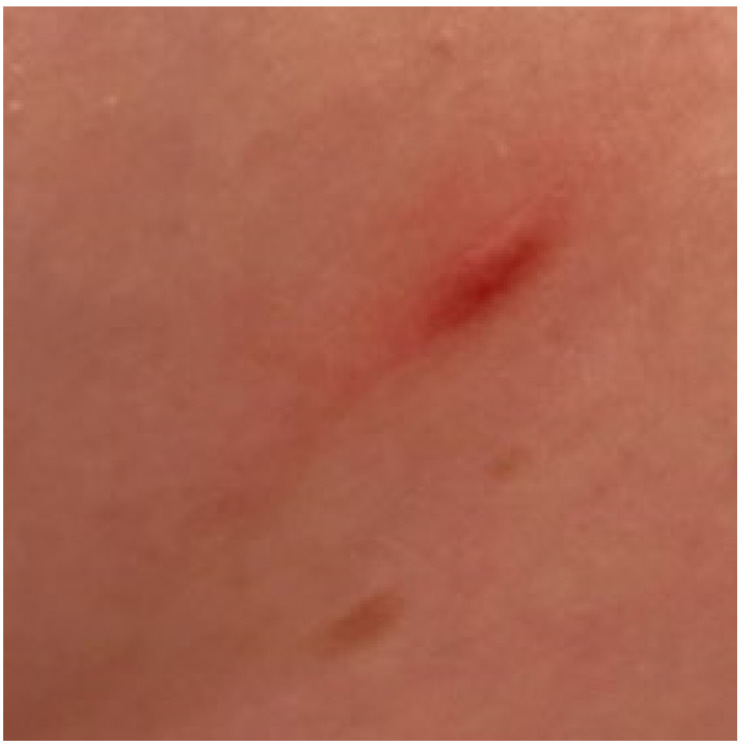
MPCM with positive Darier’s sign.

**Figure 2 diagnostics-13-03583-f002:**
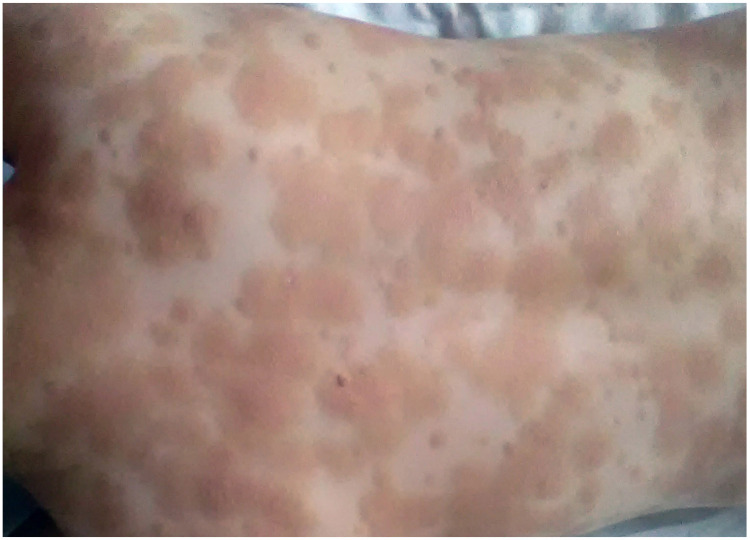
Polymorphic MPCM in an infant.

**Figure 3 diagnostics-13-03583-f003:**
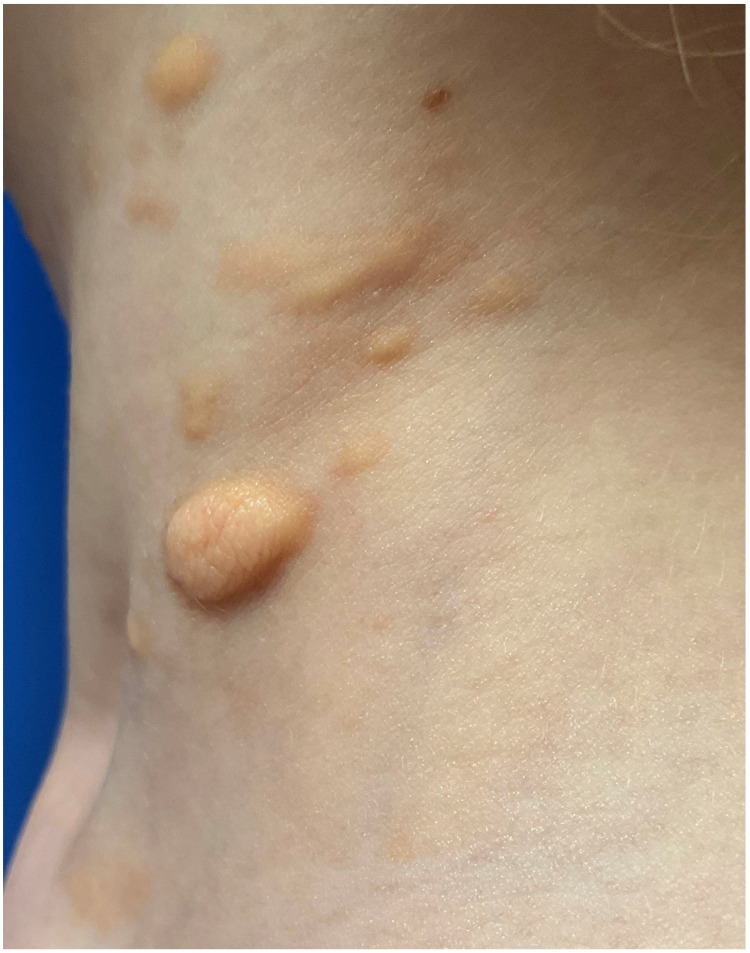
Atypical MPCM (xanthelasmoid CM) in a girl.

**Figure 4 diagnostics-13-03583-f004:**
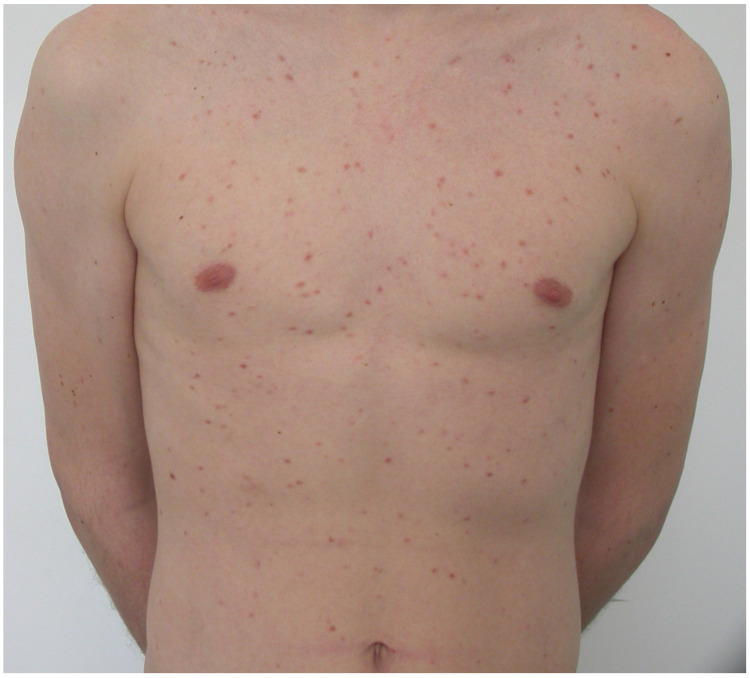
Monomorphic MPCM in a teenager.

**Figure 5 diagnostics-13-03583-f005:**
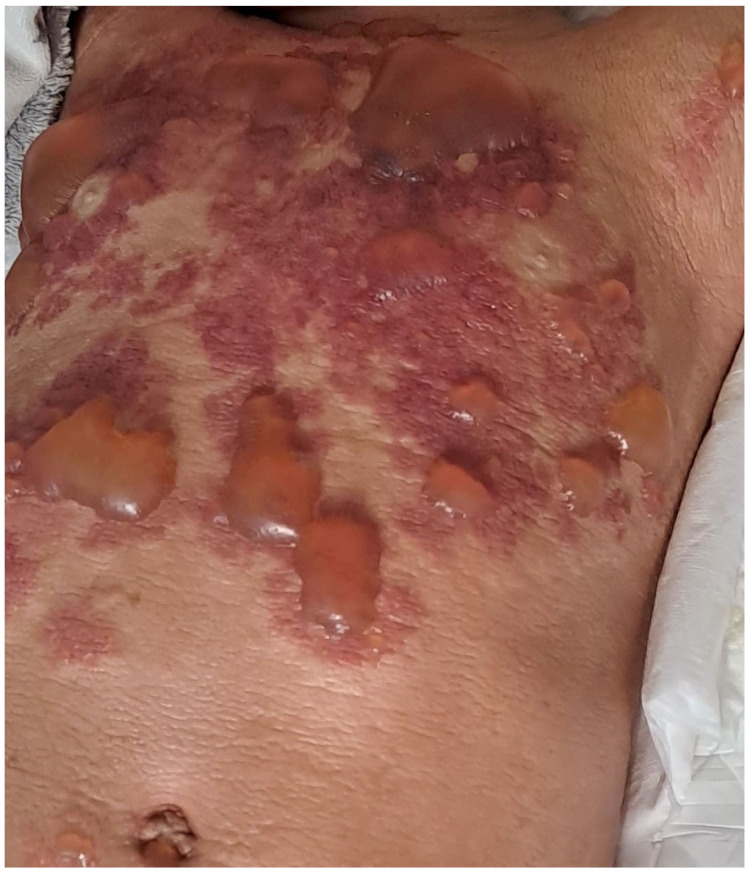
Diffuse cutaneous mastocytosis (DCM) with blistering in an infant.

**Figure 6 diagnostics-13-03583-f006:**
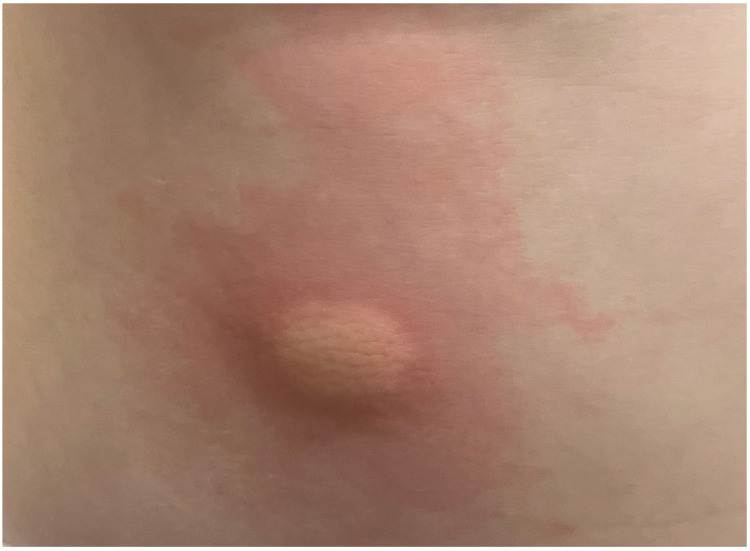
Solitary mastocytoma with positive Darier’s sign.

**Table 1 diagnostics-13-03583-t001:** Classification of mastocytosis: update 2022.

Cutaneous mastocytosis (CM) Maculopapular cutaneous mastocytosis (MPCM) ○ Monomorphic ○ Polymorphic Diffuse cutaneous mastocytosis (DCM) Mastocytoma of skin ○ Isolated mastocytoma ○ Multilocalized mastocytoma
Systemic mastocytosis (SM) Nonadvanced forms of SM • Indolent systemic mastocytosis (ISM) • Bone marrow mastocytosis (BMM) • Smoldering systemic mastocytosis (SSM) Advanced forms of SM • SM with an associated hematologic neoplasm (SM-AHN) • Aggressive SM (ASM) • Mast cell leukemia (MCL)
Mast cell sarcoma (MCS)

**Table 2 diagnostics-13-03583-t002:** Main diagnostic procedures in children with suspected or confirmed CM.

Diagnostic Procedure	Main Indications	Comments
Medical History	All children with suspected CM	Onset of the disease Presence of cutaneous and extracutaneous MC MRS and anaphylaxis Current patient’s complaints Previous and current treatment and its efficacy Concomitant diseases Family history
Physical Examination	All children with suspected CM	Assessment of morphology of skin lesions (maculopapular, large-sized, small-sized blistering, erosions, nodular, solitary, disseminated, diffuse) Assessment of the extension and distribution of skin lesions Abdominal palpation for hepatosplenomegaly Palpation of all groups of lymph nodes for unexplained lymphadenopathy The inspection of the entire skin surface
Elicitation of Darier’s sign	All children with suspected CM	Consists of erythema and urticarial swelling of the lesion when stroked Elicit carefully for mastocytoma or DCM to not provoke flushing and hypotension
PB count with differential	All children with suspected CM	PB abnormalities mainly occur in advanced SM SM may be associated with another hematologic neoplasm *FIP1L1-PDGFRA* screening if eosinophilia present
Biochemistry	All children with suspected CM	The assessment of liver and renal function Liver failure is observed in advanced SM
Basal serum tryptase level	All children with CM	An elevated level is associated with the risk of SM, anaphylaxis, and HαT The level should be assessed in correlation with the intensity of skin lesion Basal level should be determined 24–48 h after complete resolution of severe MRS or anaphylactic reaction Significantly elevated serum level is not in itself an indication of BM study
Abdominal ultrasound/CT	All children with CM	Hepatosplenomegaly and lymphadenopathy are strong indicators of SM
Skin biopsy and histological examination of lesional skin	Children with suspected CM in whom: Darier’s sign is unclear Skin lesions are not typical	The use of antibodies against tryptase and/or CD117 are recommended immunohistochemical markers of MCs
*KIT* D816V mutation in lesional skin (optionally other activating *KIT* mutations)	Children with suspected CM in whom: Histopathological examination non-diagnostic Targeted therapy with TKI is required	The presence of KIT D816V mutation or other activating KIT mutation(s) in the skin confirms the diagnosis of CM It is not a criterion of SM TKI are applied in selected children unresponsive to other therapies)
*KIT* D816V mutation in PB (optionally other activating *KIT* mutations)	Children with suspected SM	The determination of activating *KIT* mutation(s) is recommended for risk stratification of SM before deciding on a BM biopsy
BM biopsy	Children with suspected SM	BM biopsy should be considered in children with: Organomegaly Significant abnormalities in PB Significantly elevated and rising in time serum tryptase levels, Rising *KIT* D816V VAF in PB
Genetic test for *TPSAB1* copy number	Children with: Elevated serum tryptase Numerous MRS and/or anaphylaxis	Tryptase genotyping should be performed in all children with elevated serum tryptase level before deciding on a BM biopsy HαT is considered a risk factor of anaphylaxis

MRS—mediator-related symptoms; TKI—tyrosine kinase inhibitors; HαT—hereditary alfa tryptasemia; PB—peripheral blood; BM—bone marrow; VAF—variant allele frequency.

## Data Availability

Not applicable.

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
