# Peer review of "Current Challenges in the Diagnosis of Pediatric Cutaneous Mastocytosis"

_diagnostics, 2023, doi:10.3390/diagnostics13233583_

Round 1
Reviewer 1 Report
Comments and Suggestions for Authors
Dear Authors;
Well written nice article, congratulations to the author. It may be published after correcting a few following typos. I think it will contribute to the literature. Sincerely
1-It would be nice if the table 1 was arranged on a single page.
2-line 53:SM with an associated hematologic neoplasm (SM-AHN)
3- Line 200: ‘elevated serum’ much space between two words
4- Line 290: Recently, histopathological criteria of CM were validated [87]. Which organization made this validation in which year? Let's add it to the sentence
5-Line 465 and 468: please correct the typo and punto errors and remove the underscore of ‘skin’ word
Reviewer 2 Report
Comments and Suggestions for Authors
I am not sure that cutaneous mastocytosis in the children is really an haematological disease; there are data suggest that mast cells in the skin had another origine than the mast cells in the gut (Tauber, M., Basso, L., Martin, J., Bostan, L., Pinto, M. M., Thierry, G. R., Houmadi, R., Serhan, N., Loste, A., Blériot, C., Kamphuis, J. B. J., Grujic, M., Kjellén, L., Pejler, G., Paul, C., Dong, X., Galli, S. J., Reber, L. L., Ginhoux, F., … Gaudenzio, N. (2023). Landscape of mast cell populations across organs in mice and humans. Journal of Experimental Medicine, 220(10), e20230570. https://doi.org/10.1084/jem.20230570.)
